# $Z_3$-vestigial nematic order due to superconducting fluctuations in the doped topological insulators $Nb_xBi_2Se_3$ and $Cu_xBi_2Se_3$

Chang-woo Cho[1,8], Junying Shen [1,2,8], Jian Lyu [1,3], Omargeldi Atanov [1], Qianxue Chen [1], Seng Huat Lee[4,5], Yew San Hor[4], Dariusz Jakub Gawryluk [6], Ekaterina Pomjakushina [6], Marek Bartkowiak [2], Matthias Hecker[7], Jörg Schmalian[7] & Rolf Lortz[1✉]

A state of matter with a multi-component order parameter can give rise to vestigial order. In the vestigial phase, the primary order is only partially melted, leaving a remaining symmetry breaking behind, an effect driven by strong classical or quantum fluctuations. Vestigial states due to primary spin and charge-density-wave order have been discussed in iron-based and cuprate materials. Here we present the observation of a partially melted superconductivity in which pairing fluctuations condense at a separate phase transition and form a nematic state with broken $Z_3$, i.e., three-state Potts-model symmetry. Thermal expansion, specific heat and magnetization measurements of the doped topological insulators $Nb_xBi_2Se_3$ and $Cu_xBi_2Se_3$ reveal that this symmetry breaking occurs at $T_{nem} \simeq 3.8\,K$ above $T_c \simeq 3.25\,K$, along with an onset of superconducting fluctuations. Thus, before Cooper pairs establish long-range coherence at $T_c$, they fluctuate in a way that breaks the rotational invariance at $T_{nem}$ and induces a crystalline distortion.

[1] Department of Physics, The Hong Kong University of Science and Technology, Clear Water Bay, Kowloon, Hong Kong. [2] Laboratory for Neutron and Muon Instrumentation, Paul Scherrer Institute, CH-5232 Villigen PSI, Switzerland. [3] Department of Physics, Southern University of Science and Technology, Shenzhen, Guangdong 518055, China. [4] Department of Physics, Missouri University of Science and Technology, Rolla, MO 65409, USA. [5] 2D Crystal Consortium, Materials Research Institute, Pennsylvania State University, University Park, PA 16802, USA. [6] Laboratory for Multiscale Materials Experiments, Paul Scherrer Institute, CH-5232 Villigen, PSI, Switzerland. [7] Institute for Theory of Condensed Matter and Institute for Quantum Materials and Technologies, Karlsruhe Institute of Technology, Karlsruhe, Germany. [8] These authors contributed equally: Chang-woo Cho, Junying Shen. ✉email: lortz@ust.hk

Nematic electronic phases with vestigial order[1,2] are known from iron-based superconductors and cuprates, where it has been suggested that the nematic phase and the nearby spin- and charge-density wave states are not independent competing but intertwined electronic phases[1–10]. The density-wave states are the primary electronic phases and characterized by a multi-component order parameter. The nematic phase is a fluctuation-driven phase and characterized by a composite order parameter. Then the spin- or charge density-wave order melts partially, but leaves an Ising, i.e., $Z_2$-nematic state as a vestige. Vestigial order whose primary order is superconductivity has not been observed. Such partially molten superconductivity requires a material with unconventional, multi-component order parameter, and strong pairing fluctuations.

When the topological insulator $Bi_2Se_3$ is doped with electrons, e.g., by intercalation of Cu, Sr, Nb, or other metal ions in its layered structure, a superconducting state is formed[11,12]. The presence of a strong spin-orbit coupling, which also manifests itself in a topological surface state of the parent insulator[13], led to the proposal of unconventional pairing with an odd-parity symmetry and topological superconductivity[14]. The low carrier concentration, the layered structure, and the low ratio $\xi/\lambda_F$ of the superconducting coherence length and the Fermi wavelength[11,12], strongly enhance fluctuation effects. In addition, numerous experiments have shown that the superconducting state is accompanied by a spontaneous breaking of rotational symmetry with a pronounced twofold anisotropy within the $Bi_2Se_3$ basal plane[15–22]; see Ref. [23] for a recent review. The twofold symmetry can be observed in field-angle resolved experiments where a magnetic field is rotated in the plane with respect to the crystalline axes and the corresponding physical quantity (e.g., spin susceptibility, specific heat, magneto-resistance, upper critical field, magnetization, magnetic torque) is represented as a function of angle[15–23]. This behavior directly reflects the anisotropy of the superconducting state. Thus, doped $Bi_2Se_3$ is an unconventional nematic superconductor with a pairing wave function in either the two-component $E_u$ or $E_g$ point group representation, the only pairing states that spontaneously break the trifold crystal symmetry within the basal plane. The temperature dependence of the

penetration depth of Ref. [24] supports point nodes, consistent with $E_u$ odd-parity pairing.

In this article we report on high-resolution thermal expansion experiments on a superconducting mono-crystalline Nb-doped $Bi_2Se_3$ sample in combination with electrical transport, DC magnetization, and specific heat data demonstrating a $Z_3$-vestigial nematic phase with enhanced superconducting fluctuations. We have measured the linear thermal expansion in three different crystalline directions in the $Bi_2Se_3$ basal plane and observed a strong anisotropic expansion occurring at a temperature of ~0.5 K above the superconducting transition. Our high-resolution magnetization, electrical resistivity, and specific heat data—after zooming near $T_c$—show that an anomaly with increasing superconducting fluctuations occurs at the nematic transition. As we will explain below, these observations are perfectly consistent with a vestigial nematic phase of symmetry-breaking pairing fluctuations, recently predicted in Ref. [25]. This observation of a genuine symmetry breaking of pairing fluctuations above $T_c$ is qualitatively distinct from the gradual onset of order-parameter fluctuations in the disordered phase[26] or the crossover to Bose–Einstein condensation of pairs[27]. It corresponds to a sharply defined state of matter that might, e.g., undergo a separate quantum phase transition when a magnetic field is applied in the plane at low temperatures. Qualitatively similar results are obtained both on another Nb-doped $Bi_2Se_3$ single crystal and on a Cu-doped $Bi_2Se_3$ single crystal, thus, demonstrating the reproducibility and universality of the observed features.

## Results

**Magnetoresistance and nematicity.** The advantage of the Nb-doped $Bi_2Se_3$ system is that single crystals with a high superconducting volume fraction and a complete zero resistance can be found, as the results presented here show. All our bulk thermodynamic data (thermal expansion, magnetization, and specific heat) show relatively large anomalies at the superconducting transition.

Figure 1a shows magnetoresistance data recorded at 0.35 K with the magnetic field applied strictly parallel to the $Bi_2Se_3$ basal plane for different directions in the plane with respect to the

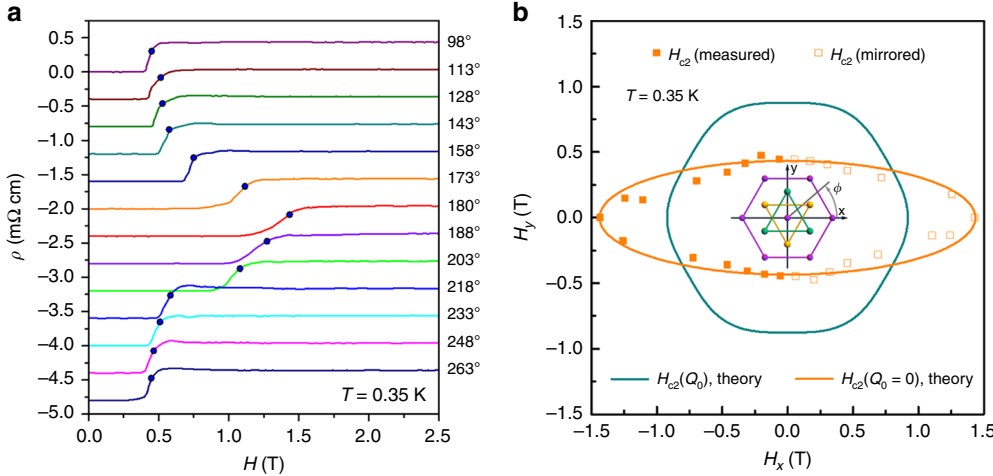

**Fig. 1 Dependence of the upper critical field $H_{c2}$ of $Nb_{0.25}Bi_2Se_3$ on the angle $\phi$ in the plane. a** Field-angle-resolved magnetoresistance data taken at 350 mK (Sample 1) for various alignments of the magnetic field in the $Bi_2Se_3$ basal plane. The additional circles mark the characteristic fields in which the magnetoresistance reaches 75% of the normal state value. The data were shifted vertically by −0.4 mΩ cm for better clarity, except for the $\phi = 98°$ data. **b** Polar plot of characteristic fields where the magnetoresistance reaches 75% of the normal state value, depending on the field direction in the $Bi_2Se_3$ basal plane. Only the full squares are real data, while the open squares are the same data shifted by 180° to better illustrate the full angular dependence. In the center, the corresponding crystal structure is added. The lines are theoretical expectations of a superconductor with trigonal symmetry without (green) and with (orange) vestigial nematic order.

trigonal crystalline axes. We determine the approximate $H_{c2}$ values from the fields in which 75% of the normal state resistance is reached and plot these values in Fig. 1b in a polar diagram as a function of angle $\phi$. A significant angular variation of $H_{c2}$ can be observed, ranging from 0.43 T at 90° where $H_{c2}$ is minimal to 1.42 T at 0° with a clear maximum of $H_{c2}$. The data show a pronounced twofold symmetry, at odds with the trifold crystalline symmetry. This is the characteristic property of nematic super-conductivity in Nb$_x$Bi$_2$Se$_3$[19,20], also known from Cu$_x$Bi$_2$Se$_3$[15,16,22] and Sr$_x$Bi$_2$Se$_3$[17,18,21]. A fit with a theoretical model for nematic SC of the form[20]

$$H_{c2}(\phi) = \frac{H_{c2}(0)}{\sqrt{\cos^2\phi + \Gamma^2\sin^2\phi}}, \qquad (1)$$

yields $\Gamma \approx 3.32\Omega$ and $H_{c2}(0) = 1.42\,\text{T}$ as a measure of the anisotropy in the basal plane. It should be noted that the normal state resistance well above the upper critical field has no variation for the different orientations of the magnetic field in the plane, indicating an isotropic normal state within the trigonal basal plane. We have previously found that the orientation of this nematic superconducting order parameter for this sample always appears to be pinned along the same of the three equivalent crystal directions in the Bi$_2$Se$_3$ basal plane, even if the sample is warmed to room temperature between different experiments[19]. The origin of this preference for a particular direction is unknown, but likely associated with microscopic details of the sample morphology, such as internal strain or microcracks (see Supplementary Discussion and Supplementary Fig. 6 for more details).

**Thermal expansion and vestigial order**. Through thermal expansion experiments, we have a highly sensitive bulk thermo-dynamic probe that is not only sensitive to the anharmonicity of phononic contributions but also to electronic degrees of freedom including nematic and superconducting order. We focus here on $\Delta L(T,H)|_\mu/L_0$ measured along different directions $\mu$ in the Bi$_2$Se$_3$ basal plane, which directly represents the change in length $\Delta L$ of the sample as a function of temperature or magnetic field, nor-malized to the length $L_0$ at ambient temperature. This quantity is directly related to the linear thermal expansion coefficient $\alpha_\mu(T,H)$ $= 1/L_0\ dL_\mu(T)/dT$. Figure 2a shows the linear thermal expansion $\Delta L|_\mu/L_0$ measured along the three directions within the Bi$_2$Se$_3$ basal plane of 90°, 155°, and 215°. All data fall perfectly on each other in the normal state, but begin to deviate gradually from each other below 3.8 K. In the following, we will refer to this characteristic temperature as $T_{nem}$, because here the onset of a twofold crystalline distortion and thus nematicity occurs. The twofold crystalline distortion with a relative length change $\Delta L/L_0 = 2 \times 10^{-7}$ amounts to a distortion of less than 0.1 femtometers within the unit cell. Still, these minute changes smaller than the size of the proton, are clearly resolvable in our measurements. The distortion is correlated with the upper critical field, with a small negative length change along 90° where the $H_{c2}$ minimum occurs (Fig. 1b), and large positive anomalies at 155° and 215°, both near the mean $H_{c2}$ value. At 3.25 K, much smaller anomalies are visible that can be identified as the superconducting transition, as the comparison with the specific heat (shown in the same graph) reveals. In the specific heat, the anomaly at $T_{nem}$ is obscured by the phonon background, but becomes visible in the temperature derivative of $C/T$ (Fig. 2b), where a small step-shaped anomaly occurs. The anomalies in $\Delta L/L_0$ at $T_{nem}$ show up as a somewhat broadened step. As a first-order derivative of the free energy, a step-like transition in $\Delta L/L_0$ is the characteristic signature of a first-order transition, while a second-order transition would appear as a kink. In distinction, the superconducting transition at $T_c$ remains the

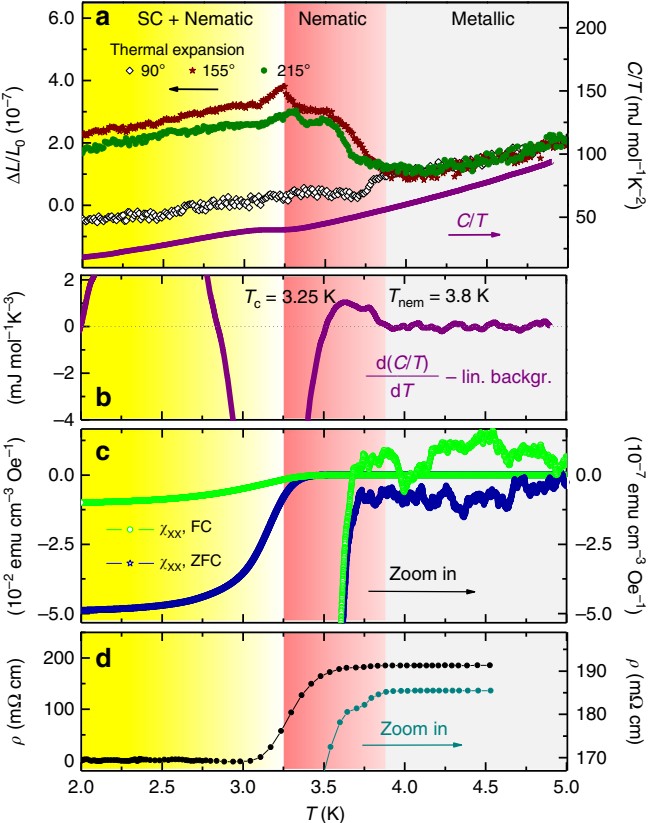

**Fig. 2 Thermal expansion, specific heat, and DC magnetization of Nb$_{0.25}$Bi$_2$Se$_3$ (Sample 1). a** Linear thermal expansion $\Delta L(T)/L_0$ (units on the left axis) measured in three directions in the Bi$_2$Se$_3$ basal plane corresponding to 90°, 155°, and 215°, together with the specific heat $C/T$ for comparison (units on the right axis). $C/T$ shows that the superconducting transition is at ~$T_c = 3.25$ K, with small kink-like anomalies occurring in the thermal expansion. A pronounced anisotropy appears in $\Delta L$ $(T)/L_0$ at a higher temperature below $T_{nem} = 3.8$ K. **b** First order derivative d$(C/T)$/d$T$ of specific heat. A linear background fitted in the range above 4 K was subtracted for reasons of clarity. The large dip centered at ~3.25 K marks $T_c$, while at $T_{nem}$ a tiny step-shaped anomaly is visible. **c** DC magnetization $M$ showing the total Meissner signal and a magnification of $10^5$ to demonstrate that the onset of superconducting fluctuations is at ~3.8 K. **d** Electrical resistivity $\rho$ showing the main superconducting transition and a magnification, which shows that the first drop in resistance due to fluctuations occurs at ~3.8 K.

standard second-order transition, as evidenced by the jump in the specific heat. Figure 2c shows the Meissner signal in the zero-field cooled and field-cooled DC magnetization, which agrees with $T_c \approx$ 3.25 K obtained from the specific heat. We also show the same data, but with a magnification of $10^5$, to illustrate that an enhanced diamagnetic response, signaling superconducting fluctuations, already sets in at $T_{nem}$. The electrical resistivity in Fig. 2d shows a similar trend with a drop in resistivity consistent with para-conductivity, i.e., superconducting fluctuations, well above the main transition.

Our X-ray diffraction results (see section "Methods") show that the doped Bi$_2$Se$_3$ phase of the $R$-3m space group is the majority phase responsible for $T_c$ at 3.25 K. As minority phases Bi$_2$Se$_3$ of space group P-3m1 and NbBiSe$_3$ were found. The latter occupy far too little volume to explain such large anomalies in thermal expansion. From this, we conclude that the observed crystalline distortion below $T_{nem} = 3.8$ K is caused by a transition separate from the main superconducting transition, but linked to the

**Fig. 3 Lattice distortion and real space image of the superconducting order parameter.** Purple dashed lines below $T_{nem}$ indicate the high-temperature atomic positions. The lattice distortion is strongly exaggerated. Arrows in the nematic phase indicate the directions of the thermal expansion measurements. In the latter, strong superconducting fluctuations break the discrete lattice symmetry without broken $U(1)$ symmetry and superconducting coherence. A globally coherent superconducting state only sets in at the superconducting transition temperature $T_c$.

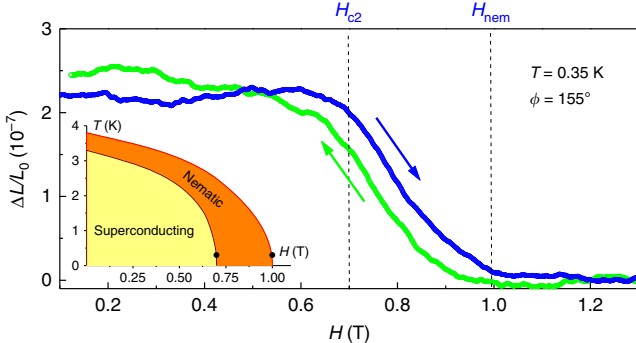

**Fig. 4 Magnetostriction $\Delta L(T)/L_0$ as a function of the magnetic field.** The data (Sample 1) was measured along the 155° direction at a fixed temperature of 350 mK. The crystalline distortion is removed at $H_{nem} = 1\,T$, well above the resistively determined $H_{c2}$, which coincides with the kink at 0.7 T for this field orientation (marked as $H_{c2}$). The magnetic field was applied parallel to the measured sample direction. A weak linear normal state background was subtracted for clarity. The inset shows the magnetic phase diagram.

occurrence of superconducting fluctuations, which cause a weak Meissner effect and decrease in resistivity below this temperature.

**Two stage transition.** Our data thus show that nematic superconductivity occurs in the form of a two-stage transition, see also Fig. 3 for an illustration. The distortion forms near the higher onset temperature $T_{nem}$, where superconducting fluctuations in the magnetization and the first-order derivative of the specific heat are visible. The superconducting transition occurs at a lower temperature $T_c$, which corresponds to the formation of a global phase-coherent superconducting state. The signs and magnitudes of the length changes for the three measuring directions are consistent with the indicated distortions, see "Methods" section.

It could be argued that thermal expansion reveals a separate structural transition at $T_{nem}$ that has nothing to do with superconductivity[28]. Such a sequence of independent or competing transitions would be allowed within the Landau theory of phase transitions. Obviously, the anomaly in the diamagnetic response at $T_{nem}$ is already strong evidence that this is not the case. Furthermore, in Fig. 4 we show magnetostriction data for the 155° direction, along which we observed the greatest change in length in Fig. 2a. Here, $\Delta L/L_0$ was measured at a fixed temperature of 350 mK as a function of the magnetic field. A

broad step-like transition occurs with a total length change of $\Delta L/L_0 \approx 0.22 \times 10^{-7}$ with onset at ~1.0 T. Figure 1 shows that the resistively determined $H_{c2}$ for this direction occurs at 0.7 T. Magnetostriction shows that the crystalline distortion is constant up to this field where a kink occurs, then it is gradually removed up to 1.0 T. The overall anomaly represents a broad step in $\Delta L/L_0$, which is the expected characteristics of a first-order transition. A small hysteresis can be seen in the data measured upon sweeping the field up and down. Experimental artifacts as source for the hysteresis have been carefully excluded, especially since the field scanning speed was kept very slow at 0.02 T/min, which typically does not cause any hysteresis-like effects in reversible samples. Given the layered structure of the sample with the field aligned in parallel, this hysteresis is most likely a consequence of flux pinning effects, which are typically strong at such a low temperature. Therefore, the hysteresis should not necessarily be regarded as evidence of a first-order $H_{c2}$ nature. The field-induced length change at low temperature shown in Fig. 4 corresponds to the temperature-induced length change at zero field shown in Fig. 2a. This observation provides further evidence that the nematic distortion is closely linked to the superconducting state, with a separate nematic transition $H_{nem}$ occurring above the main superconducting $H_{c2}$ transition, as shown in Fig. 4.

Further data on thermal expansion measured on a second sample of the same batch are shown in Supplementary Figs. 1 and 2 and are discussed in Supplementary Note 1. They show a similar behavior, although the crystalline distortion occurring below $T_{nem}$ is weaker due to a multi-domain structure. For this sample we have also measured thermal expansion in fixed magnetic fields, and it can be seen that $T_{nem}$ is suppressed by the magnetic field together with $T_c$, further confirming that the two transitions are closely related. We also show similar data of a $Cu_{0.2}Bi_2Se_3$ single crystalline sample (Sample 3) in Supplementary Fig. 3–5 and the data are discussed in Supplementary Note 2.

**Discussion**
The two thermodynamic properties, the specific heat and the linear thermal expansion coefficient $\alpha_\mu(T) = 1/L_0\ dL_\mu(T)/dT$ are closely related in the vicinity of a phase transition through the Clausius Clapeyron (2) and Ehrenfest (3) relation for first and second order phase transitions, respectively. The proportionality is the uniaxial pressure dependence $dT_c/dp_\mu$ of the transition ($T_c$ is the critical temperature at which the phase transition occurs, $p_\mu$ is the uniaxial pressure applied along a certain crystalline direction $\mu$, $V_{mol}$ is the molar volume, $\Delta S$ is the jump in entropy at a

first order transition and $\Delta C_p$ is the jump in the specific heat at constant pressure at a second order phase transition).

$$\frac{dT_c}{dp_\mu} = \frac{\left(\frac{\Delta L}{L_0}\right)_\mu \cdot V_{mol}}{\Delta S}, \tag{2}$$

$$\frac{dT_c}{dp_\mu} = \frac{\Delta \alpha_\mu \cdot T_c \cdot V_{mol}}{\Delta C_p}. \tag{3}$$

A large anomaly in thermal expansion and a small anomaly in specific heat means in both cases that $T_{nem}$ is strongly dependent on uniaxial pressure and the electronic nematic order is strongly coupled to the crystalline lattice. The strong crystalline distortion observed here using linear thermal expansion therefore means that the nematic transition is strongly dependent on pressure or strain. Such a behavior can also be observed, for example, in iron based superconductors, where a nematic transition occurs in the vicinity to a spin density wave transition and causes large anomalies in thermal expansion[29].

Our findings can be explained in terms of vestigial order due to superconducting fluctuations. In fact, recently it has been suggested that such vestigial order should emerge from the superconducting phase in doped $Bi_2Se_3$[25]. On the one hand, the superconducting order parameter of either the $E_g$ or the $E_u$ representation has two components[14,30–32]

$$\left(\Delta_x, \Delta_y\right) = \Delta_0 e^{i\varphi}(\cos\theta, \sin\theta), \tag{4}$$

that are characterized by the overall amplitude $\Delta_0$, the global $U(1)$ phase $\varphi$ and three distinct values of the angle $\theta = \left\{\frac{\pi}{6}, \frac{\pi}{2}, \frac{5\pi}{6}\right\}$ that select a specific crystalline axis. Superconducting fluctuations will then induce a phase transition to a vestigial nematic state at a temperature $T_{nem}$ above $T_c$. While superconductivity is signaled by a finite expectation value of $\Delta_x$ and/or $\Delta_y$, the nematic phase is characterized by a finite expectation value of the composite order parameter

$$\mathbf{Q} = \begin{pmatrix} |\Delta_x|^2 - |\Delta_y|^2 & \Delta_x^*\Delta_y + \Delta_y^*\Delta_x \\ \Delta_x^*\Delta_y + \Delta_y^*\Delta_x & |\Delta_y|^2 - |\Delta_x|^2 \end{pmatrix}. \tag{5}$$

Upon increasing the temperature, superconducting fluctuations continue to break the rotational symmetry, even after restoration of global $U(1)$ symmetry at $T_c$. The composite order parameter $\mathbf{Q}_{\mu\nu}$ is made up of combinations of the superconducting order parameter, similar to charge-4e superconductivity proposed within the context of pair-density wave order in cuprate superconductors[33,34] or proton–electron superconducting condensate in liquid hydrogen[35]. As a traceless second-rank tensor, $\langle \mathbf{Q}_{\mu\nu} \rangle = Q_0 \left(n_\mu n_\nu - \frac{1}{2}\delta_{\mu\nu}\right)$ behaves, however, like a nematic order parameter with director $\mathbf{n} = (\cos\theta, \sin\theta)$[36] and strongly couples to the strain tensor $\boldsymbol{\varepsilon}_{\mu\nu}$ via $\kappa \text{tr}(\mathbf{Q}\varepsilon)$ with nemato-elastic coupling constant $\kappa$. A nonzero $Q_0$ then induces a lattice distortion $\boldsymbol{\varepsilon}_{\mu\nu} \propto \kappa \mathbf{Q}_{\mu\nu}$, see Fig. 3. Thus, the lattice can be utilized to detect this unconventional electronic order. The point group analysis further yields a first-order transition at $T_{nem}$ into a state with $Q_0 \neq 0$, since it is in the three-state Potts model, i.e., the $Z_3$ universality class. The superconducting transition continues to be of second order, all in agreement with our experimental findings. In Fig. 5a we show the nematic ($Q_0$) and superconducting ($\Delta_0$) order parameters and in Fig. 5b the diamagnetic susceptibility obtained within the theory of Ref. [25]. The susceptibility is compared with the data of Fig. 2c, where the logarithmic axis is used to illustrate the rapid growth of diamagnetic fluctuations below $T_{nem}$. While the in-plane anisotropy of the

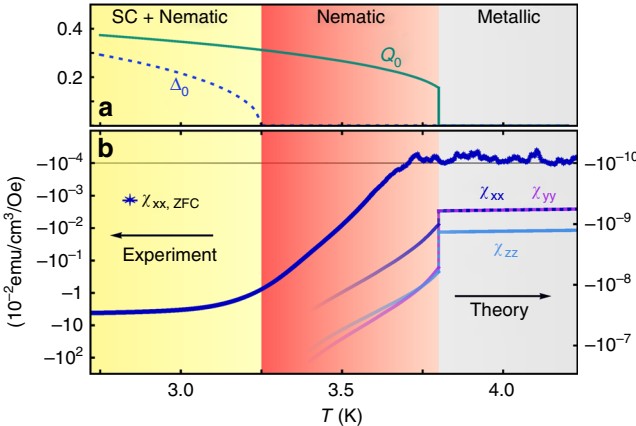

**Fig. 5 Superconducting and nematic order parameters and diamagnetic susceptibility. a** The two intertwined order parameters $\Delta_0$ (superconducting) and $Q_0$ (nematic). **b** The anisotropic diamagnetic susceptibilities $\chi_{xx}$, $\chi_{yy}$ and $\chi_{zz}$ together with the experimental magnetization data on a log-linear scale to compare with the theory.

susceptibility is only finite below the nematic transition, the crystal symmetry allows $\chi_{zz}$ to be distinct already above $T_{nem}$. The magnitude of the out-of-plane anisotropy is determined by the ratio of the electron velocities in the corresponding directions. The anisotropy of $H_{c2}(\phi)$ shown in Fig. 1b (orange line) was also obtained within the same theory and is compared with the behavior without nematic order ($Q_0 = 0$) where $H_{c2}$ should have sixfold symmetry[24,25]. Without nematic phase above $T_c$, the superconducting order parameter directly at $H_{c2}(\phi)$ is infinitesimal and no twofold rotational symmetry breaking should be visible, in clear contrast to experimental observations. In Ref. [31], the twofold symmetric behavior of $H_{c2}(\phi)$ only occurred after an additional symmetry-breaking strain was added. Vestigial nematic order offers a natural explanation for this strain field.

After this work was completed, we learned about Ref. [37], where a twofold symmetry breaking above $T_c$ is reported. Our results agree with those of Ref. [37] and make evident that the high temperature phase is separated by an actual first order transition where superconducting fluctuations are enhanced. Furthermore, Refs. [38,39] reported on the control of nematic superconductivity by uniaxial strain, which is consistent with our observation of a coupling of the nematic order parameter to the crystal lattice.

To summarize, our data demonstrate that a separate nematic transition occurs in the doped topological insulator $Nb_{0.25}Bi_2Se_3$ at $T_{nem} = 3.8$ K, i.e., about 0.5 K above $T_c$, with a distinct crystalline distortion occurring in the $Bi_2Se_3$ basal plane. $T_{nem}$ coincides with the onset temperature of superconducting fluctuations. The direction of the crystalline distortion is correlated with the direction of the twofold symmetry of the superconducting order parameter and is removed together with the superconductivity at or near the upper critical field $H_{c2}$. The two transitions are thus interconnected. Our observations are perfectly consistent with vestigial nematic order and a sequential restoration of $U(1)$ and rotational symmetry. The new nematic phase is a state of matter in which Cooper pairs have lost their off-diagonal long-range order, yet fluctuate in a way that breaks the rotational symmetry of the crystalline lattice.

## Methods

**Sample characterization.** The monocrystalline $Nb_{0.25}Bi_2Se_3$ sample used in this study was selected because of its particularly large $T_c$ anomalies in the specific heat, which indicates a high superconducting volume fraction, and because of its particularly large nematic in-plane $H_{c2}$ anisotropy[19]. Our previous work also demonstrated that it forms one large nematic domain comprising ~90% of the superconducting volume fraction in which the nematic order parameter is pinned

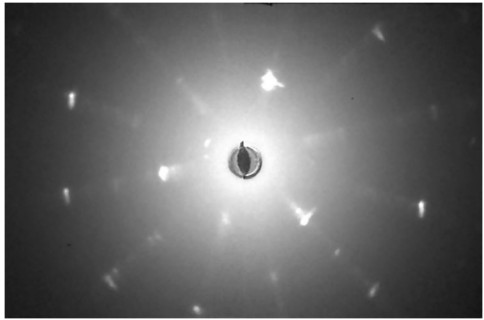

**Fig. 6 Laue X-ray diffraction image of the $Nb_xBi_2Se_3$ single crystal (Sample 1).** This data allowed us to define the crystalline orientation in the $Bi_2Se_3$ basal plane.

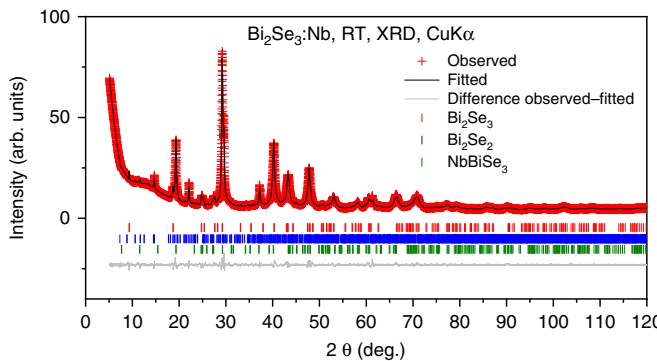

**Fig. 7 X-ray diffraction pattern for $Nb_xBi_2Se_3$ (Sample 1).** The room temperature CuKα radiation pattern is shown by the red crosses. The black line corresponds to the best fit from the Rietveld refinement analysis. Lower vertical marks denote the Bragg peak positions of $Bi_2Se_3$ (red), $Bi_2Se_2$ (blue), and $NbBiSe_3$ (green). The bottom, gray line represents the difference between experimental and calculated points.

in one crystalline direction, while the remaining 10% is due to a minority domain in which the orientation is rotated by 60°. Data from a second monocrystalline $Nb_{0.25}Bi_2Se_3$ sample with broader transition anomalies and somewhat less nematic anisotropy are shown in Supplementary Figs. 1 and 2. We also show data of a $Cu_{0.2}Bi_2Se_3$ single crystalline sample (Sample 3) in Supplementary Fig. 3–5. These samples show a qualitatively similar behavior as Sample1.

A laboratory X-ray Laue equipped with CCD camera (Photonic Science) was used to characterize the crystal quality and to determine the crystalline directions in the basal $Bi_2Se_3$ plane. The Laue images (Fig. 6) were taken on the shiny surface of the sample after cleaving off a thin layer with shooting X-ray beam along the $c$-axis, revealing the crystal orientation and proving the hexagonal structure. We took data on different spots on the sample surface and found that the change in the crystalline direction was less than 0.2° over a distance of 0.8 mm, proving the sufficiently good single crystalline quality of the sample.

Powder X-Ray Diffraction pattern was collected at room temperature in the Bragg-Brentano geometry using a Bruker AXS D8 Advance diffractometer equipped with a Ni-filtered Cu Kα radiation and a 1D LynxEye PSD detector (Fig. 7). The reflections were indexed to $Bi_2Se_3$ (SG $R-3m$; No 166) as a main phase. In addition, some minority phases were found as following: $Bi_2Se_2$ and $NbBiSe_3$ with corresponding space groups: $P-3m1$ (No 164) and $P2_12_12_1$ (No 19), respectively. This demonstrates that, while most of the Nb must be intercalated between the $Bi_2Se_3$ layers, some Nb is incorporated into the layers on the Bi sites. This agrees with literature data[40]. The Rietveld refinement[41] of the diffraction patterns was done by the package FULLPROF SUITE[42] (version July-2019) using a previously determined instrument resolution function (based on the small line width polycrystalline sample $Na_2Ca_3Al_2F_{14}$ measurements[43]). Refined parameters were: scale factor, zero displacement, lattice parameters, atomic positions, isotropic Debye–Waller factors, and peak shape parameters as a Thompson-Cox-Hastings pseudo-Voigt function. Determined lattice parameters of the rhombohedral $Bi_2Se_3$ are equal $a = b = 4.1854(4)$ Å, and $c = 28.4633(7)$ Å. Because of the habit of the powdered crystal, a preferred orientation as a March-Dollase multi-axial phenomenological model was implemented in the analysis.

**Experimental techniques**. The high-resolution linear thermal expansion was measured with a capacitive technique using a dilatometer, in which the sample is pressed by a fine screw mechanism against a cantilever forming one of the two plates of a capacitor. A change in the sample length leads to a change in the separation of the capacitor plates, which can be determined with a General Radio 1615A capacitance bridge in combination with a Stanford Research SR830 digital lock-in amplifier. Before the experiments we have carefully checked that the empty dilatometer does not show any anomalies in the temperature range of interest (Supplementary Fig. 7). We have measured the thermal expansion as a function of temperature in three different directions within the basal $Bi_2Se_3$ plane (90°, 155°, and 215°). 0° is the $a$ direction normal to the mirror plane and corresponds to the magnetic-field direction in the plane providing the maximum upper critical field, while 90° corresponds to the $a^*$ direction parallel to the mirror plane. The crystalline directions have been obtained by Laue X-ray diffraction as explained above. The other directions were chosen as representative of other characteristic directions in the plane, but largely dictated by the crystal shape, which allowed a stable mounting of the sample only for certain directions. All directions were within 5° from the three different $a^*$ directions. For technical reasons, all data was taken at a slow rate of 0.02 K/min upon increasing temperature. Figure 8 shows a photograph of the mounted crystal for the three different orientations in the dilatometer. The dilatometer[44] was well characterized using separate calibration measurements: e.g., a measurement with a Cu sample of 1 mm length of the same material as the dilatometer body shows only a very small temperature dependence without significant anomalies in the temperature range of interest. The absolute value of the change in length was calibrated using a 2 mm long undoped silicon sample, which gave a linear thermal expansion coefficient that agrees well with literature[45] (Supplementary Fig. 8).

The specific heat was measured with a home-made calorimeter, which can be used either in AC modulated temperature mode or in long relaxation mode. The long relaxation mode provides high accuracy in the absolute value of 1% precision, while the AC mode provides high relative resolution with a high density of data points of 1000 points per K. The data presented in this letter has been acquired using the AC technique, but the absolute value has been calibrated using the relaxation technique.

DC magnetization was measured using a commercial Quantum Design Vibrating Sample SQUID magnetometer and the electrical resistance was measured using a standard 4-probe technique with a Keithley 6221 AC current source combined with a SR830 digital lock-in amplifier. For the latter, a low temperature piezo rotator was used to precisely align the magnetic field along the different crystalline directions and to study the $H_{c2}$ anisotropy that reflects the nematic superconductivity. The rotator allowed milli-degree precision for relative changes in orientation. However, a systematic error of less than 5° can occur with respect to the measured crystalline axes.

**Theory: q-state Potts model**. A $q$-state Potts model describes a spin-like variable $s = 1, 2, ..., q$ that can attain $q$ different values. In our case $q = 3$ and the three values label the three axes of the crystal along which one assumes a local displacement. The energy of two neighboring variables $s$ and $s'$ that are the same is then lower than for distinct variables, i.e., the bond energy goes like $-\frac{1}{2}J_0(q\delta_{s,s'} - 1)$. In general, the model has to be distinguished from a $q$-states clock model where an angle $\varphi \in [0, 2\pi]$ can take $q$ distinct values $\varphi_s = \frac{2\pi s}{q}$ and two sites interact via an energy proportional to $-J_0\cos(\varphi_s - \varphi_{s'})$. For $q = 3$ both models are equivalent though. Using the Potts language one can define the fraction $n_s$ of the lattice in the $s$-th state, which implies $n_1 + n_2 + n_3 = 1$. At high temperatures one expects $m_s = n_s - \frac{1}{3}$ to have zero expectation value for all $s$. Below the transition temperature, one of the $m_s$ becomes positive and the two others negative. Because of the condition on the sum of the $n_s$ the three $m_s$ are not independent: $m_1 + m_2 + m_3 = 0$. Furthermore, the free energy of the system must be symmetric under $m_s \leftrightarrow m_{s'}$. These two conditions lead to the unique free energy expansion up to quartic order

$$f = r\sum_{s=1}^{3} m_s^2 - v m_1 m_2 m_3 + u\left(\sum_{s=1}^{3} m_s^2\right)^2. \qquad (6)$$

Since there are really only two independent variables one can introduce the parametrization $m_1 = \frac{2}{\sqrt{3}}Q_1$ and $m_{2,3} = \frac{1}{2} - \frac{1}{\sqrt{3}}Q_1 \pm Q_3$. The expansion in terms of the $Q_{1,2}$ leads up to constants to

$$f = r\left(Q_1^2 + Q_2^2\right) - v'Q_1\left(Q_1^2 - 3Q_2^2\right) + u\left(Q_1^2 + Q_2^2\right)^2 \qquad (7)$$

This is precisely the free energy expansion obtained in Ref. [25] in terms of the quadrupolar order parameter of Eq. (2) with $Q_1 = |\Delta_x|^2 - |\Delta_y|^2$ and $Q_2 = \Delta_x^*\Delta_y + \Delta_y^*\Delta_x$. This demonstrates that the problem at hand is indeed in the universality class of the $q = 3$ Potts or clock models.

The sketch of the distorted unit cell in Fig. 3 is deduced from the linear coupling term $\kappa$tr$(\mathbf{Q}\varepsilon)$ between the strain tensor and the composite order parameter. Thus, the nematic order parameter acts in the same way as an applied external stress field in the $E_g$ symmetry channel and hence distorts the unit cell. Moreover, the linear coupling term does not entail a change of the unit cell volume, which is assumed to be unaltered in the following. Figure 9 shows the relative length changes in the three directions 90°, 155°, and 215° caused by an $E_g$ unit cell deformation as a function of the lattice parameter $a/a_0$. For the strongly exaggerated value $a/a_0 = 1.1$

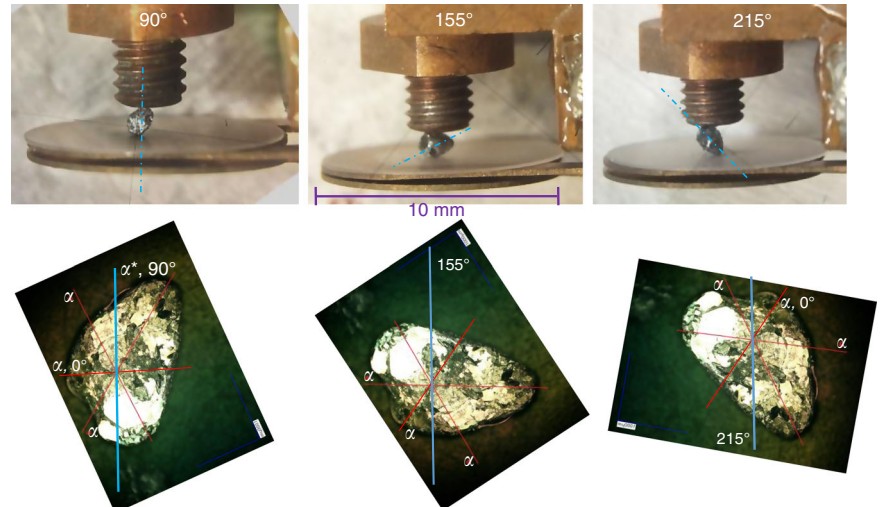

**Fig. 8 Photographs of Sample 1 mounted in the capacitive dilatometer.** The three panels show the $Nb_xBi_2Se_3$ sample mounted along the three measured directions within the $Bi_2Se_3$ basal plane. The dotted lines mark the 90° direction, which corresponds to one of the crystalline $a^\star$ directions (parallel to the mirror plane). The lower row of images illustrates the corresponding crystalline orientations. The red lines mark the crystalline $a$ directions (perpendicular to the mirror plane) as determined by Laue X-ray diffraction. The blue lines mark the measured directions.

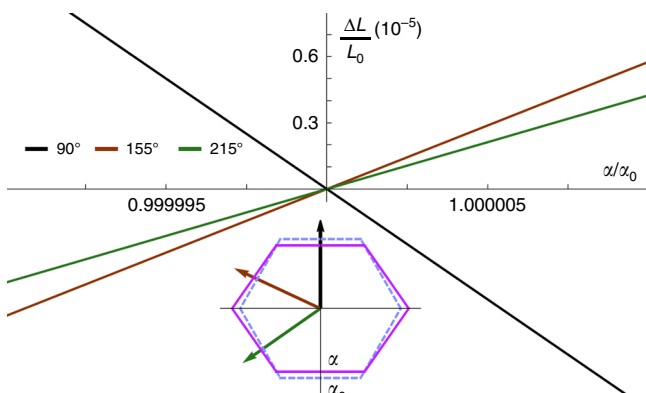

**Fig. 9 Relative length changes in the three directions 90°, 155°, and 215°.** The length changes are caused by an $E_g$ unit cell deformation as a function of the lattice parameter $a/a_0$. The inset shows a distorted hexagon for an exaggerated value of $a/a_0 = 1.1$.

we also show the corresponding distorted hexagon in the inset, which is quantitatively similar to the distorted unit cell in Fig. 3. While the computed relative length changes qualitatively capture the measured thermal expansion behavior, the calculated magnitude of the 90° direction is slightly larger than that observed in experiment when compared with the other two directions. This could be due to the higher-order coupling that gives rise to a change of the unit cell volume at $T_{nem}$.

## Data availability

The experimental data supporting the findings of this work are available at https://doi.org/10.4121/uuid:8f2eed77-3db3-4d07-965a-d4900f5ff22d.

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

## Acknowledgements

We thank U. Lampe for the technical support and acknowledge enlightening discussions with I. R. Fischer, C. Meingast, K. T. Law, and K. Willa. This work was supported by grants from the Research Grants Council of the Hong Kong Special Administrative Region, China (GRF-16302018, SBI17SC14, IEG16SC03). J.S. was supported by the Gordon and Betty Moore Foundation's EPiQS Initiative through Grant GBMF4302 and GBMF8686 while visiting the Geballe Laboratory for Advanced Materials at Stanford University. J.S. also acknowledges support by the German Research Foundation (DFG) through the Collaborative Research Center CRC TRR 288 "Elastic Tuning and Response of Electronic Quantum Phases of Matter", project B01. Y.S.H. acknowledges the support from the NSF-DMR 1255607.

## Author contributions

This work was initiated by R.L., J.Y.S. carried out the magneto-transport measurements with help of M.B., C.w.C., Q.C., and R.L. carried out the thermal expansion measurements with help of O.A., C.w.C., and R.L. carried out the specific heat and DC magnetization measurements with help of J.L.; the $Nb_xBi_2Se_3$ single crystal samples were provided by S.H.L. and Y.S.H., the $Cu_xBi_2Se_3$ single crystal sample was provided by E.P., M.H. and J.S. provided the theoretical simulations and further theoretical support. The X-ray characterization of the sample was done by D.J.G. and J.Y.S., the manuscript was prepared by R.L. and J.S. with help of C.w.C. and M.H. All authors were involved in discussions and contributed to the manuscript.

## Competing interests

The authors declare no competing interests.
