## [Peer Review File · Nature Communications]

Reviewers' comments:

Reviewer #1 (Remarks to the Author):

The manuscript by Cho et al., reports observations of the fluctuation phenomena in $\text{Nb}_x\text{Bi}_2\text{Se}_3$ crystal above the superconducting transition temperature. These fluctuations are believed to be nematic, i.e. crystal symmetry breaking. And, if they are indeed observed, this would be an important achievement in the field of unusual superconductivity, systems with multicomponent order parameter, and surely deserve the publication in Nature Communications.

While the theoretical part of the paper and idea are excellent, the experimental data did not convince me that the mentioned fluctuation-induced symmetry breaking indeed occurs. Please find my comments below:

1. The authors demonstrate deviations of various physical observables from the normal behavior slightly above T_c . This should be an indicator of the fluctuation regime (vestigial nematic order). However, the same observations might be explained if the sample is nonuniform and has some small inclusions with elevated T_c . Therefore some additional data are needed to rule out sample non-uniformity: structural data (see below), and, probably, measurements in magnetic field. Magnetic field should affect superconductivity and vestigial nematic features in a different manner, that could probably help to disentangle fluctuation effects from non-uniformity. Indeed, Nb has a lot of superconducting compounds, admixture of some side fraction could make the observations misleading.

2. Structural data.

Of course, a photo of the sample (given in the manuscript) is not enough to say anything about its chemical uniformity, structural quality, or even crystalline orientation.

X-ray diffraction studies are necessary.

In the Methods section the authors state:

"The detailed growth method and characterization of

$\text{Nb}_{0.25}\text{Bi}_2\text{Se}_3$ in the mono-crystalline form can be found in Ref. 19."

I looked through Ref. 19 (Shen, J., et al., npj Quantum Mater. 2, 59 (2017).), and found almost the same statement in the Method section:

"The detailed growth method and characterization of

$\text{Nb}_{0.25}\text{Bi}_2\text{Se}_3$ in the single crystalline form can be found in refs. 24,25."

This means, at least, that the statement in the manuscript is false. However, I decided to look through the Refs. 24 and 25 from the Ref. [19].

Ref 24 (Asaba, T. et al. Phys. Rev. X 7, 011009 (2017).) also did not contain any X-ray diffraction data and sent the reader to Ref 25.

Ref 25 is an unpublished preprint (<https://arxiv.org/abs/1512.03519>) where X-ray diffraction data are also absent.

I conclude therefore that there are no published structural studies for these crystals. Not only anything can be told about the crystal quality, uniformity, but even 3-fold rotational axis of the material is not confirmed. This is a very serious argument against publication of this paper.

3. Thermal expansion. The paper reports low-temperature thermal expansion coefficients of 10^{-5} 1/K. It is not clear for me why the values are so huge, comparable to the room temperature value.

Indeed, it is a textbook knowledge that thermal expansion originates from unharmonicity of the atomic potential. E.g., for most of the cryogenic materials the thermal expansion below liquid nitrogen temperature is known to be vanishing. There should be a reason and explanation of such huge mechanical effects.

I might suspect that the observed values of thermal expansion are artificial and related to the dilatometer itself.

Moreover, the jump, related to superconducting transition (Fig. 4) might be also due to more trivial reasons, e.g. force, acting on superconducting sample in magnetic field.

Therefore, I believe that before implementation to $\text{Nb}_x\text{Bi}_2\text{Se}_3$, the reliability of the dilatometer should be tested preliminary with a known superconductors and materials of known heat expansion coefficients.

4. Related to 3. The data obtained with a single sample are always suspicious. It easily might be that the sample consists of two blocks (see discussion in Ref. 23) and an isotropic direction in Fig. 2a is related to the direction of the inter-block boundary.

=====

Unfortunately, the above drawbacks make the paper unsuitable for publication.

Reviewer #2 (Remarks to the Author):

This manuscript has reported high-resolution measurements of thermal expansion, specific heat, and magnetization of the doped topological insulator $\text{Nb}_x\text{Bi}_2\text{Se}_3$, which is known to be superconducting below $T_c=3.25$ K. The authors claimed that the experimental findings suggest the existence of a Z_3 -vestigial nematic order caused by the superconducting fluctuation within a narrow range of temperatures $T_c < T < T_{nem}$ where T_{nem} is roughly 3.8 K. Such a vestigial nematic order was theoretically predicted in Ref.[25] by two of the authors.

Vestigial order is a new concept that has recently been proposed and investigated in the context of unconventional superconductors, including cuprates and iron-based superconductors. It might provide a useful generic framework to understand the very complicated phase diagram of these compounds. It is interesting that such a phenomenon can also occur in $\text{Nb}_x\text{Bi}_2\text{Se}_3$, a widely studied candidate for nematic superconductor. The thermodynamic data reported in this manuscript provide convincing evidence for the existence of an intermediate, fluctuation-induced nematic order above T_c . I believe that these results would be of general interest to the theorists and experimentalists working on unconventional superconductors.

The results presented in this manuscript are original, and the conclusion is sharp and interesting. The paper is well written, and can be easily understood by the readers. The Abstract is very clear,

containing the most important conclusion. The measurements are performed by using standard methods. Other groups would be able to repeat the experiments following the Methods part.

I would like to recommend a publication of this manuscript on Nature Communications.

Reviewer #3 (Remarks to the Author):

I have reviewed the manuscript NCOMMS-19-16698 titled "Z3-vestigial nematic order due to superconducting fluctuations in the doped topological insulator $\text{NbxBi}_2\text{Se}_3$ " by Chang-woo Cho et al., submitted to Nature Communications.

In this manuscript, the authors studied properties of Nb-doped Bi_2Se_3 superconductor by measuring the resistivity, magnetic susceptibility, specific heat, and most importantly thermal expansion. The authors report an interesting finding: a nematic structural distortion occurring at a temperature slightly above the superconducting critical temperature T_c . This structural transition temperature, T_{nem} , seems to coincide with the onset of the superconducting fluctuation. By comparing the experimental findings and theoretical expectation, the authors conclude that the observed structural transition is driven by the fluctuation of the nematic superconductivity, namely "vestigial order" of the nematic superconductivity.

The findings reported is, if it is intrinsic, quite interesting, opening a new type of physics behind the nematic superconductivity. Vestigial order is related also to a wide range of other interesting systems such as cuprates or iron-based superconductors. In addition, the present material is believed to be a bulk topological superconductor. Thus, this new study should be interesting to a wide community of superconductivity and topological materials science. Nevertheless, I have several concerns on the results. In particular, the provided information is not sufficient to fully convince me that the observed T_{nem} is intrinsic, as discussed in detail below.

To conclude, the present manuscript is worth for publication in Nature Communications, but only after my concerns mentioned below are fully clarified.

Below, I list my point-by-point comments:

[1] I have concern whether the observed structural transition at T_{nem} is intrinsic or not. Thus, the authors should clarify the issues listed below.

(1) First of all, experimental evidence that the observed signal is not due to apparatus background should be provided. I am particularly concerned that T_{nem} is close to the superconducting transition temperature of Sn, which is probably used somewhere in the apparatus as solder. Thus, I request the authors to provide data of various control experiments; such as dilatometer response measured without any samples, and dilatometer response measured with known standard materials, to confirm that any anomaly is absent near T_{nem} without $\text{NbxBi}_2\text{Se}_3$.

(2) The authors should discuss the value of the observed thermal expansion coefficient with those reported in previous literatures (on $\text{NbxBi}_2\text{Se}_3$; or pure Bi_2Se_3 , if the data of former are not available).

(3) The authors should discuss the reproducibility of the observed behavior. I recommend the authors to measure at least one more sample to confirm that the observed behavior is intrinsic.

(4) The authors should explain whether the $\Delta-L/L$ curves for the temperature-up sweep and temperature-down sweep overlap each other, besides possible hysteresis at T_{nem} due to the proposed first-order nature of the transition.

(5) In relation to (3) and (4), if the order at T_{nem} is truly the realization of the Z_3 Potts model, the lattice distortion below T_{nem} may differ in different cooling processes. Explain this was the case or not. If not, add a brief discussion on possible mechanisms of such nematicity "pinning".

(6) Are there a possibility of multiple nematic domains within the sample? If yes, can existence of the domains affect the interpretation?

(7) Explain the definitions and used values of L_0 and T_0 . And discuss whether the conclusion is independent of the choices of the values of L_0 or T_0 .

[2] Explanation of the magneto-striction data (Fig.4) is not enough. Answer the questions listed below.

(1) Is the structural change (at H_{nem}) still located above the superconducting H_{c2} (i.e. $H_{nem} > H_{c2}$) or is H_{nem} equal to H_{c2} ?

(2) Is the transition at H_{nem} still a first-order transition or not?

(3) Does the hysteresis in Fig.4 indicate the first-order nature or is it just within experimental uncertainties?

(4) Can the authors exclude the possibility of dominance of magnetic torque signal, which can be significant if one apply field off from high-symmetric directions?

[3] Although the transition at T_{nem} is claimed to be triggered by superconducting fluctuations, the change of the lattice parameter is clear and substantial. Then, a question arises: can the anomaly in the specific heat be such tiny? The authors should discuss the size of the anomaly in the specific heat more carefully, based on thermodynamic consideration and/or theoretical calculation. It might be also helpful to compare with nematic phase transitions observed in other nematic electron systems.

[4] In the manuscript, it is not clear whether the measured sample(s) are identical, or they are not identical but taken from the same batch, or they are taken from different batches. Please add explanation.

[5] For the in-field measurements (Fig.1 and Fig.4), explain the accuracy of the field alignment with respect to the crystalline axes. Then, discuss possible influence of the field misalignment to the observed data.

[6] If T_{nem} is also a onset of superconducting fluctuation, T_{nem} should be clearly seen in the temperature dependence of resistivity measured at zero field. Add an explanation whether the

resistivity also onsets at T_{nem} . If not, the interpretation by the authors becomes a bit questionable.

[7] Explain at which field the magnetic susceptibility shown in Fig.2c was measured. Also, explain whether the demagnetization correction was made when calculating the susceptibility. By the way, I prefer to use the volume susceptibility (emu/cc/Oe) rather than the mass susceptibility (emu/g/Oe) when discussing the superconducting shielding effect.

[8] Add a bit more explanation what the "Z3 Potts model" is, for a broad audience of Nature Communications.

[9] Add a brief explanation of the origin of the finite calculated susceptibility in the normal state (above T_{nem}) in Fig.5.

[10] This is a minor point but the vertical axis of Fig.5 is a bit strange: log scale should not contain zero. Consider to use another better way of presentation, or explain the definition of the vertical axis more carefully, to avoid confusion of non-specialists. Actually the shoulder-like structure in the experimental data at around 3.6 K (-10^{-4} emu/g/Oe) is probably an artifact, originating from the change of scales between a log scale to linear scale.

Response to Reviewer #1

We would like to thank reviewer 1 for his careful reading and his numerous valuable suggestions which have revealed the weaknesses of our manuscript. In the following, we will address his comments one by one.

Comment: While the theoretical part of the paper and idea are excellent, the experimental data did not convince me that the mentioned fluctuation-induced symmetry breaking indeed occurs.

Our reply: *In the past months we have made great efforts to conduct additional experiments that support our interpretation. We now have the additional data and hope that they meet the expectations of the reviewer.*

Comment: 1. The authors demonstrate deviations of various physical observables from the normal

behavior slightly above T_c . This should be an indicator of the fluctuation regime (vestigial nematic order). However, the same observations might be explained if the sample is non-uniform and has some small inclusions with elevated T_c .

Our reply: *We agree that structural data and more detailed sample characterization was needed (which we have now done, see below). In addition, we would like to point out that the thermal expansion shows a pronounced crystalline distortion well above T_c , while a much smaller anomaly is seen at T_c . The large anomalies in thermal expansion cannot be explained by small inclusions with increased T_c , especially since thermal expansion is a bulk thermodynamic method. In the case of a non-uniformity, all quantities including the thermal expansion should show a large anomaly at T_c with broadening to higher temperatures or some smaller anomalies at higher temperatures if minority phases were responsible. Our X-ray diffraction results show that the doped Bi_2Se_3 phase of $R\text{-}\bar{3}m$ space group is the majority phase responsible for the T_c at 3.3K. Minority phases found were Bi_2Se_3 of space group $P\text{-}\bar{3}m1$ and NBiSe_3 . The latter occupy far too little volume to explain such large anomalies in the thermal expansion. We have included a brief discussion in the manuscript to point this out (page 7, line 1-7), in addition to the detailed results of the X-ray characterization we added to the Methods section (page 19, line 13 – page 20, line 11).*

Comment: Therefore, some additional data are needed to rule out sample non-uniformity: structural data (see below), and, probably, measurements in magnetic field. Magnetic field should affect superconductivity and vestigial nematic features in a different manner, that could probably help to disentangle fluctuation effects from non-uniformity.

Our reply: *As mentioned above, we have performed a detailed structural analysis and included it in the Methods section. Although Nb-doping certainly induces disorder, we find no evidence for an extended minority phase that could explain such large anomalies in the thermal expansion, which is a bulk thermodynamic property. Furthermore, different samples show a qualitatively similar behavior in the thermal expansion, including a Cu-doped sample (see our new Supplementary Materials).*

We have added data in magnetic fields obtained from measurements on a second sample. The upper transition at T_{nem} is also suppressed by the magnetic field, showing that the nematic order is strongly linked to the superconducting order, which is expected by the theory. These additional data are also contained in the supplementary materials.

Comment: Indeed, Nb has a lot of superconducting compounds, admixture of some side fraction could make the observations misleading.

Our reply: We do not observe a large proportion of other Nb compounds that could explain the large anomalies in the thermal expansion at T_{nem} , which is a bulk thermodynamic quantity. Any small admixture of a secondary phase would be expected to only show up as a small contribution. The only secondary phases we observed were Bi_2Se_3 of space group $P-3m1$ and $NbBiSe_3$. In addition, we present data of Cu-doped Bi_2Se_3 , which also shows a transition above T_c even at a slightly higher temperature above 4 K. Elemental Cu is not a superconductor.

Comment: 2. Structural data.

Of course, a photo of the sample (given in the manuscript) is not enough to say anything about its chemical uniformity, structural quality, or even crystalline orientation. X-ray diffraction studies are necessary.

Our reply: *An X-ray diffraction analysis was performed and is included in the Methods section of the revised manuscript (page 19, line 13 – page 20, line 11).*

Comment: In the Methods section the authors state:

"The detailed growth method and characterization of $\text{Nb}_{0.25}\text{Bi}_2\text{Se}_3$ in the mono-crystalline form can be found in Ref. 19."

I looked through Ref. 19 (Shen, J., et al., npj Quantum Mater. 2, 59 (2017).), and found almost the same statement in the Method section:

"The detailed growth method and characterization of $\text{Nb}_{0.25}\text{Bi}_2\text{Se}_3$ in the single crystalline form can be found in refs. 24,25."

This means, at least, that the statement in the manuscript is false. However, I decided to look through the Refs. 24 and 25 from the Ref. [19].

Ref 24 (Asaba, T. et al. Phys. Rev. X 7, 011009 (2017).) also did not contain any X-ray diffraction data and sent the reader to Ref 25.

Ref 25 is an unpublished preprint (<https://arxiv.org/abs/1512.03519>) where X-ray diffraction data are also absent.

I conclude therefore that there are no published structural studies for these crystals. Not only anything can be told about the crystal quality, uniformity, but even 3-fold rotational axis of the material is not confirmed. This is a very serious argument against publication of this paper.

Our reply: *We apologize, this was a misunderstanding between us and our collaborator who supplied the single crystals to us. We agree that such detailed characterization is necessary. Dr. Dariusz Jakub Gawryluk, an expert for such X-ray characterization at PSI in Switzerland, kindly helped us to perform such a detailed characterization on the single crystal used in this study, which, as mentioned above, is included in the Methods section. This also led to an extended list of coauthors of the manuscript.*

Comment: 3. Thermal expansion. The paper reports low-temperature thermal expansion coefficients of 10^{-5} 1/K. It is not clear for me why the values are so huge, comparable to the room temperature value. Indeed, it is a textbook knowledge that thermal expansion originates from unharmonicity of the atomic potential. E.g., for most of the cryogenic materials the thermal expansion below liquid nitrogen temperature is known to be vanishing. There should be a reason and explanation of such huge mechanical effects.

Our reply: *First of all, we would like to apologize. The absolute values of the thermal expansion have been overestimated by a factor of 10 due to a mechanical problem in the switches of our General Radio Capacitance bridge, which indicates the range of the capacitance measurement. This has been corrected. The size of the anomalies is now correct. In addition, we would like to point out that thermal expansion is a bulk thermodynamic property similar to specific heat, and that in addition to the phonon contribution, there is also a contribution from conduction electrons. The observed anomalies are largely due to electronic degrees of freedom related to the volume dependence of the Sommerfeld constant, which can*

be relatively large at low temperatures. We have added a more detailed description of the thermal expansion to the revised manuscript (page 5, line 15-21).

Comment: I might suspect that the observed values of thermal expansion are artificial and related to the dilatometer itself.

Our reply: The dilatometer itself shows no anomaly in this temperature range. Below is a comparison between the new raw data of Sample 2 (measured at $\theta=25$ degrees, blue) with an empty measurement (red):

We also do not see how anomalies caused by the dilatometer itself could lead to anomalies around 4 K, which are different along different directions in the plane, and with an anisotropy that correlates with the anisotropy of the superconducting order parameter. Note that in the revised manuscript we now have data from 3 different samples (two of which are included in the supplementary materials), all of which show a qualitatively similar behavior, with the length of the sample expanding along the 0-degree direction and shrinking along the 90-degree direction. The dilatometer has been in use in our laboratory since 2010 and has been extensively tested and calibrated and used e.g. on iron-based superconductors: *Physica C* **539**, 30 (2017).

Comment: Moreover, the jump, related to superconducting transition (Fig. 4) might be also due to more trivial reasons, e.g. force, acting on superconducting sample in magnetic field.

Our reply: This is a reasonable concern if one only considers the data shown in Fig. 4. However, for the measurements shown in Figure 2, no magnetic field was applied, which excludes such a force as explanation of our observation. In addition, we also applied a magnetic field when we measured the magnetostriction shown in in Figure 4. Still the size of the anomaly at H_{c2} is perfectly consistent with the deformation that occurs at T_{nem} in zero field. This would clearly not be the case if it was due to a

torque effect. Furthermore, the deformation at T_{nem} occurs in a phase without superconducting long range coherence, where macroscopic screening currents that could generate such a force or torque are not present.

Comment: Therefore, I believe that before implementation to $Nb_xBi_2Se_3$, the reliability of the dilatometer should be tested preliminary with a known superconductors and materials of known heat expansion coefficients.

Our reply: See our publication in *Physica C* **539**, 30 (2017) for a measurement on another superconductor that is well in agreement with literature data (*Physical Review B* 86, 094521 (2012)). We also show below data measured on silicon (here we plot the linear thermal expansion coefficient $\alpha = 1/L_0 dL/dT$, which is the temperature derivative of $\Delta L/L_0$), compared to literature data showing good agreement. This measurement served us to test the absolute value of the thermal expansion of our dilatometer.

Comment: 4. Related to 3. The data obtained with a single sample are always suspicious. It easily might be that the sample consists of two blocks (see discussion in Ref. 23) and an isotropic direction in Fig. 2a is related to the direction of the inter-block boundary.

Our reply: We measured a second $Nb_{0.25}Bi_2Se_3$ single crystalline sample. While the observed crystalline distortion is weaker, it shows qualitatively the same characteristics at T_{nem} (see Supplementary Materials). The lower distortion is due to the fact that this second sample has two minority domains with different orientation of the nematic order parameter in the plane, which we know from specific heat experiments in a magnetic field. In addition, we included data from a $Cu_{0.2}Bi_2Se_3$ single crystalline sample, which also shows qualitatively similar behavior. Regarding the data used in the main manuscript, it was measured on exactly the same sample that was used for our previous publication in *npj Quantum Materials* 2, 59 (2017), where, from a fit to the angular dependence of the upper critical field, we estimated that it consists to 90% of a single domain, with a 10% volume fraction minority phase rotated 60 degrees from the majority phase. The overall behavior is thus dominated by one large nematic

domain. Our X-ray diffraction characterization of this sample found that it is a good single crystal with a variation in crystalline orientation of not more than 0.2 degrees as measured at different locations on the crystal. A detailed description of the nature of Sample 1 is included at the begin of the Methods section of the revised manuscript (page 19, line 3-12).

Comment: Unfortunately, the above drawbacks make the paper unsuitable for publication.

Our reply: *We hope that our new data can convince reviewer 1 about the validity of our approach.*

Response to Reviewer #2

Comment: The results presented in this manuscript are original, and the conclusion is sharp and interesting. The paper is well written, and can be easily understood by the readers. The Abstract is very clear, containing the most important conclusion. The measurements are performed by using standard methods. Other groups would be able to repeat the experiments following the Methods part. I would like to recommend publication of this manuscript on Nature Communications.

Our reply: *We thank the reviewer for this positive evaluation!*

Response to Reviewer #3

Comment: The findings reported is, if it is intrinsic, quite interesting, opening a new type of physics behind the nematic superconductivity. Vestigial order is related also to a wide range of other interesting systems such as cuprates or iron-based superconductors. In addition, the present material is believed to be a bulk topological superconductor. Thus, this new study should be interesting to a wide community of superconductivity and topological materials science. Nevertheless, I have several concerns on the results. In particular, the provided information is not sufficient to fully convince me that the observed T_{nem} is intrinsic, as discussed in detail below.

To conclude, the present manuscript is worth for publication in Nature Communications, but only after my concerns mentioned below are fully clarified.

Our reply: *We thank the reviewer for this positive evaluation. We have invested the last few months to conduct additional experiments in response to his comments. We hope that the additional data meet the expectations of the reviewer.*

Comment: (1) First of all, experimental evidence that the observed signal is not due to apparatus background should be provided. I am particularly concerned that T_{nem} is close to the superconducting transition temperature of Sn, which is probably used somewhere in the apparatus as solder. Thus, I request the authors to provide data of various control experiments; such as dilatometer response measured without any samples, and dilatometer response measured with known standard materials, to confirm that any anomaly is absent near T_{nem} without $NbxBi_2Se_3$.

Our reply: Below is a comparison between the new raw data from Sample 2 (measured at $\theta=35^\circ$ degrees, blue) with an empty measurement (red). The empty dilatometer shows no anomalies in the temperature range of interest. This is a standard test that we always perform when we have a new instrument.

Comment: (2) The authors should discuss the value of the observed thermal expansion coefficient with those reported in previous literatures (on $NbxBi_2Se_3$; or pure Bi_2Se_3 , if the data of former are not available).

Our reply: We could not find in the literature high-resolution reference data for Bi_2Se_3 or related compounds in this low temperature range. The lowest temperature data we found for pure Bi_2Se_3 extended down to ~ 10 K (Features of low-temperature thermal expansion of n-type Bi_2Se_3 single crystals in magnetic field, *Bulletin of the Lebedev Physics Institute* 44(11):324-326v(2017)). The value of thermal expansion $\Delta L/L$ at 10 K was about 2×10^{-6} , which agrees quite well with an extrapolation of our low-temperature data up to this temperature. However, we have some calibration data. Below we show a measurement that we performed for calibration purposes on a silicon single crystal in comparison with literature data. Here we have plotted the linear thermal expansion coefficient $\alpha = 1/L dL/dT$, which shows good agreement with the literature data and well reflects the temperature range in which silicon has a negative expansion.

Comment: (3) The authors should discuss the reproducibility of the observed behavior. I recommend the authors to measure at least one more sample to confirm that the observed behavior is intrinsic.

Our reply: We have measured a second $Nb_xBi_2Se_3$ sample that shows the same structural deformation at T_{nem} . The deformation is smaller, but this is to be expected, given that we know that this sample has two minority domains due to twinning effects where the nematic anisotropy is along different crystal directions. This can be seen, for example, in the specific heat measured with an ac technique at constant temperature during a field sweep, where we observed two additional smaller H_{c2} anomalies associated with the two minority domains. However, the overall behavior is very similar. The new data were incorporated in the supplementary materials. We also have some first data on $Cu_{0.2}Bi_2Se_3$, where we have the same anisotropy below T_{nem} , which is slightly higher at ~ 4.2 K, well above the T_c at 3.3 K. Again, the splitting of the crystalline axes is weaker, which in this case is likely due to the lower charge density in Cu doped Bi_2Se_3 . The following is a comparison of the data of 3 different samples (Sample 1 & Sample 2: $Nb_{0.25}Bi_2Se_3$, Sample 3: $Cu_{0.2}Bi_2Se_3$).

Comment: (4) The authors should explain whether the $\Delta L/L$ curves for the temperature-up sweep and temperature-down sweep overlap each other, besides possible hysteresis at T_{nem} due to the proposed first-order nature of the transition.

Our reply: *Unfortunately, for technical reasons related to our temperature control in the cryostat, we can currently only measure upon heating. Small hysteresis effects can be seen in our magnetostriction experiments, for which we have data measured during up and down sweeps of the magnetic field.*

Comment: (5) In relation to (3) and (4), if the order at T_{nem} is truly the realization of the Z₃ Potts model, the lattice distortion below T_{nem} may differ in different cooling processes. Explain this was the case or not. If not, add a brief discussion on possible mechanisms of such nematicity "pinning".

Our reply: *We have not observed any significant differences in measurements after different cooling processes. In addition, it is known from the literature and agrees with our own observations that the nematic order parameter is firmly pinned in a certain crystalline direction and always orientates itself in the same direction, even when the sample is warmed to room temperature. Although the reason for this is still unknown, it is most likely due to the exact sample morphology, such as internal stress or microcracks, which influence the direction of the nematic order. We added a short discussion in our manuscript (page 5, line 9-14) as well as to the supplementary materials (section "Nematic transition and internal strain fields" on page 5).*

Comment: (6) Are there a possibility of multiple nematic domains within the sample? If yes, can existence of the domains affect the interpretation?

Our reply: *In our revised manuscript we now show data from two different $Nb_{0.25}Bi_2Se_3$ samples (data of the second ($Nb_{0.25}Bi_2Se_3$) and a third sample ($Cu_{0.2}Bi_2Se_3$) are included in the supplements). Sample 1 is almost a monodomain sample. It is the same sample that was used in our previous publication (npj Quantum Materials 2, 59 (2017)). In the plot of H_{c2} there is only a very small additional bump indicating a 10% volume contribution of a minority domain rotated by 60 degrees. Sample 2 has two larger minority*

domains, so that the upper critical field transition in the magnetoresistance is broadened. In the field dependence of the specific heat we can resolve the upper critical field anomalies of these two minority domains. As a result, the overall crystalline distortion measured in the thermal expansion is smaller but qualitatively similar. We added an explanation into the Methods section of our manuscript (page 19, line 3-12).

Comment: (7) Explain the definitions and used values of L_0 and T_0 . And discuss whether the conclusion is independent of the choices of the values of L_0 or T_0 .

Our reply: We have chosen L_0 as sample length at room temperature. Since the variation of the sample length between room temperature and 300 K is in the permill range, the choice of L_0 does not have a significant influence on our data. $\Delta L(T)$ is measured directly, while α is determined by differentiation of $\Delta L(T)/dL$. Therefore the choice of T_0 has no effect on our data. Typically, $T=0$ would be selected. We found it more useful to define the relationship between $\Delta L/L_0$ and α by the differential, where $\alpha(T) = 1/L dL/dT$, since it is determined experimentally in this way. We have updated the definition of the thermal expansion in the revised manuscript accordingly (page 5, line 15-21).

Comment: [2] Explanation of the magneto-striction data (Fig.4) is not enough. Answer the questions listed below.

(1) Is the structural change (at H_{nem}) still located above the superconducting H_{c2} (i.e. $H_{nem} > H_{c2}$) or is H_{nem} equal to H_{c2} ?

Our reply: At the low temperature at which these data were recorded, there is only a broad step-like anomaly with a length that is gradually approaching the normal state. In order to address the question raised by the referee we compare this observation with our new magnetoresistance data. This allows us to determine H_{c2} much more precisely. The resistive H_{c2} appears at the lower onset of the broad step in the magnetostriction (marked as H_{c2} in the new Figure 5 shown below), while the step above corresponds to a gradual removal of the nematic distortion, indicating that H_{nem} is indeed located above the superconducting H_{c2} . Note that here we mark the highest field below which a deviation from the normal state appears as H_{nem} (which makes sense because this is where the deformation begins), while for a first order transition the true thermodynamic critical field would be better chosen as the midpoint of the broad step. Both designations lead, however, to the same conclusion that H_{nem} and H_{c2} are distinct. We have marked this accordingly in the revised figure 4 in our manuscript (see figure below) and added a few sentences discussing this (page 7, line 22 – page 8, line 5).

Comment: (2) Is the transition at H_{nem} still a first-order transition or not?

Our reply: As for the large width of this step-shaped feature in magnetostriction, it is not easy to interpret, but a step-shaped feature in $\Delta L/L_0$ is consistent with a first order nature. It is expected from the Potts model that the H_{c2} transition remains first order at all temperatures and this agrees with the step-like feature we observe. This is mentioned in the new paragraph we have added on page 7, line22.

Comment: (3) Does the hysteresis in Fig.4 indicate the first-order nature or is it just within experimental uncertainties?

Our reply: The experimental uncertainties are much smaller, so yes, we do observe a real hysteresis effect. With a superconductor, however, one must be careful when interpreting a hysteresis if a measurement is performed at such a low temperature as a function of the magnetic field. Due to the lack of thermal energy, the energy barriers for flux pinning are very large. This alone can cause hysteresis effects that are not necessarily related to a first-order nature of the upper critical field transition. To investigate hysteresis effects, data would be required that are measured for different fields during heating and cooling. For technical reasons, however, measurements during cooling in our cryostat are not possible. We have mentioned this in the new paragraph we have added on page 7, line22.

Comment: (4) Can the authors exclude the possibility of dominance of magnetic torque signal, which can be significant if one applies field off from high-symmetric directions?

Our reply: The essential data in our manuscript presented in Fig. 2 are zero-field data for which no torque can be present. As for the magnetostriction experiments, we also perform torque experiments in our laboratory with a similar cantilever setup, but the cantilevers we use for torque are 10 times thinner. Our springs, which hold the sample in the dilatometer, are quite strong. The total change in length in magnetostriction is always equal to what we observe at T_c in zero field, so it is very unlikely that we

observe a torque artefact. To further address this issue we included new data where we have measurements at constant magnetic fields that show how the transition shifts to lower temperature, but with a similar shape as in zero field. The magnetostriction anomalies for all orientations correlate exactly with the zero field anomalies, indicating the same origin. There is no torque in zero field, suggesting that a torque is not the origin of the observed anomalies.

Comment: [3] Although the transition at T_{nem} is claimed to be triggered by superconducting fluctuations, the change of the lattice parameter is clear and substantial. Then, a question arises: can the anomaly in the specific heat be such tiny? The authors should discuss the size of the anomaly in the specific heat more carefully, based on thermodynamic consideration and/or theoretical calculation. It might be also helpful to compare with nematic phase transitions observed in other nematic electron systems.

Our reply: *The specific heat and the linear thermal expansion coefficient $\alpha = 1/L dL/dT$ are closely related in the vicinity of a phase transition by the Clausius Clapeyron and Ehrenfest relation for first and second order transitions respectively, where the proportionality is the uniaxial pressure dependence of the transition. A large anomaly in the thermal expansion and a small anomaly in the specific heat means that T_{nem} is strongly dependent on uniaxial pressure and the electronic nematic order is strongly coupled to the crystalline lattice. If the nematic transition is coupled to the crystalline lattice and causes a crystalline distortion - and we observe this - it means that it is strongly pressure dependent and therefore it is expected to observe much larger anomalies in thermal expansion than in specific heat. Such a behavior can also be observed, for example, in iron based superconductors, where a nematic transition takes place in the vicinity to a spin density wave transition causing large anomalies in thermal expansion (Physical Review B 86, 094521 (2012)). We have added a discussion in the revised manuscript (page 8, line 16 - page 9, line 10).*

Comment: [4] In the manuscript, it is not clear whether the measured sample(s) are identical, or they are not identical but taken from the same batch, or they are taken from different batches. Please add

explanation.

Our reply: *All data in the main text of the manuscript were measured on the same single crystalline sample ("Sample 1"). We have included data from a second $Nb_{0.25}Bi_2Se_3$ sample ("Sample 2") in the supplementary materials, and from a $Cu_{0.2}Bi_2Se_3$ sample ("Sample 3"). This is now stated more clearly at the begin of the Methods section (page 19, line 3-12).*

Comment: [5] For the in-field measurements (Fig.1 and Fig.4), explain the accuracy of the field alignment with respect to the crystalline axes. Then, discuss possible influence of the field misalignment to the observed data.

Our reply: *Magnetoresistance was measured using a piezo rotary stage with milli-degree precision for relative changes in orientation. However, a systematic error of less than 5 degrees can occur with respect to the measured crystalline axes. Anything larger would be easily detected by the symmetry of the obtained H_{c2} curve. For the thermal expansion we had to rely on photos of the sample in the dilatometer to measure the orientation, which we believe is possible with an accuracy of 5 degrees. This should have no influence on the interpretation. We have included this in the Methods section of the revised manuscript (page 21, line 3 and page 21, line 21).*

Comment: [6] If T_{nem} is also a onset of superconducting fluctuation, T_{nem} should be clearly seen in the temperature dependence of resistivity measured at zero field. Add an explanation whether the resistivity also onsets at T_{nem} . If not, the interpretation by the authors becomes a bit questionable.

Our reply: *Yes, we have such resistance data and we have included it in the revised Fig. 2d. The resistance onset occurs indeed at T_{nem} .*

Comment: [7] Explain at which field the magnetic susceptibility shown in Fig.2c was measured. Also, explain whether the demagnetization correction was made when calculating the susceptibility. By the way, I prefer to use the volume susceptibility (emu/cc/Oe) rather than the mass susceptibility (emu/g/Oe) when discussing the superconducting shielding effect.

Our reply: *The data were measured in 2 Oe. No demagnetization correction was made, partly because this is not really necessary in such a small field, partly because the sample was a thin platelet oriented parallel to the applied field so that the demagnetization factor can be neglected. Concerning the units, we have updated the magnetization graphs accordingly, thank you very much for the suggestion!*

Comment: [8] Add a bit more explanation what the "Z3 Potts model" is, for a broad audience of Nature Communications.

Our reply: *A paragraph describing the Z3 Potts model has been added to the Methods section (page 22 & 23).*

Comment: [9] Add a brief explanation of the origin of the finite calculated susceptibility in the normal state (above T_{nem}) in Fig.5.

Our reply: *This constant offset is very small and is within the resolution limit of the device (note that Fig. 5 is a logarithmic plot). We have added new data (both in Fig. 2c and 5) and you see in the zoomed in data that the ZFC and FC data have slightly different offset. This occurs close to the resolution limit and may originate from very few Fe impurities, which are difficult to avoid since our magnetometer is not in a clean room environment.*

Comment: [10] This is a minor point but the vertical axis of Fig.5 is a bit strange: log scale should not contain zero. Consider to use another better way of presentation, or explain the definition of the vertical axis more carefully, to avoid confusion of non-specialists. Actually the shoulder-like structure in the experimental data at around 3.6 K (-10^{-4} emu/g/Oe) is probably an artifact, originating from the change of scales between a log scale to linear scale.

Our reply: *We apologize. This indeed does not make sense and we have revised the presentation. We now use the new ZFC data, which does not show this tiny positive offset.*

We hope that our revised manuscript, the additional data and the explanations we provide will meet the reviewer's expectations.

REVIEWERS' COMMENTS:

Reviewer #1 (Remarks to the Author):

Dear editors and authors,

First of all I am happy indeed that several improvements were done to this paper:

- scale of the elongation was corrected following my comment
- $\text{Cu}_x\text{Bi}_2\text{Se}_3$ was added
- logic was improved

Now the paper is scientifically sound.

I have only two minor issues prior to acceptance:

1. Comment on hysteresis in Figure 4.

Indeed, there are other reasons for the hysteresis:

- some magnetic parts are inside the setup those are magnetized with field (in this case hysteresis should be temperature-independent)
- field sweep rate is too high that may cause cooling or heating of the paramagnetic parts, especially at low temperatures (this can be excluded from the direction of the hysteresis loop)
- there is a time lag between correct recording of magnetic field and the measurement (field sweep rate must be indicated).

2. Concerning the strain dependence of the nematic properties. There are two recent research those address this issue in AxBi_2Se_3 materials:

<https://arxiv.org/abs/1910.03252> and <https://doi.org/10.1103/PhysRevB.100.224509>

The authors should somehow comment on relation of their results with these studies.

Reviewer #3 (Remarks to the Author):

I have reviewed the manuscript NCOMMS-19-16698A with a modified title "Z3-vestigial nematic order due to superconducting fluctuations in the doped topological insulators $\text{NbxBi}_2\text{Se}_3$ and $\text{Cu}_x\text{Bi}_2\text{Se}_3$ " by Chang-woo Cho et al., re-submitted to Nature Communications.

After the first-round review, the authors substantially revised the manuscript, with lots of additional experimental data. One main concern pointed out by me and Reviewer #1, namely the reliability of the dilatometer, is now resolved. (It now turns out that there was a factor 10 mistake in the strain data in the previous manuscript.) With these substantial revision, I now agree with the publication of the manuscript in Nature Communications.

I have several additional comments that could help improving the paper.

[A] Concerning my comments 1 and [1](2) in the previous round, it would be better to include some figures in the response letter to the manuscript (probably in Supplemental Information); the one comparing raw $\Delta L/L_0$ data of Sample 2 and the empty dilatometer, and the one comparing measured thermal expansion coefficient (α) of silicon with the literature data. These two figures are quite convincing and should be shown to the readers.

[B] Concerning my comment [1](4), please add a short sentence on the sweep direction in Method.

[C] Concerning my comment [9], I asked about the CALCULATED susceptibility, but the authors response was on the MEASURED susceptibility. I'm curious why there is a substantial anisotropy between χ_{xx}/χ_{yy} and χ_{zz} in Fig.5. It would be nice if the authors put a short explanation on this in the final version.

[D] In the abstract (line 27), "partially melted superconductor" should be "partially melted superconductivity" or "partially melted superconducting order parameter".

[E] In the introduction (line 42), "Vestigial order, where the primary order is superconductivity, has not ..." should be "Vestigial order whose the primary order is superconductivity has not ...". Here, restrictive clause should be used. Thus, commas should be removed.

We would like to thank all reviewers for their dedication to help us to improve our article!

In the following we respond to all their remarks and suggestions and explain how we accounted for them in our revised manuscript.

Responses to Reviewer #1

Comment 1:

1. Comment on hysteresis in Figure 4.

Indeed, there are other reasons for the hysteresis:

- some magnetic parts are inside the setup those are magnetized with field (in this case hysteresis should be temperature-independent)
- field sweep rate is too high that may cause cooling or heating of the paramagnetic parts, especially at low temperatures (this can be excluded from the direction of the hysteresis loop)
- there is a time lag between correct recording of magnetic field and the measurement (field sweep rate must be indicated).

Our response:

We are grateful for the detailed thoughts and suggestions regarding the origin of the hysteresis in our field sweep experiment. The field sweep rate during our experiment was only 0.02 T/min, and we measure the magnetic field directly with a Hall probe to avoid artifacts due to time lags. There is also no magnetic material in the vicinity of our dilatometer. The probe and the dilatometer are made of either high-purity Cu or CuBe. In fact, we know from various experiments that with reversible materials (no flux pinning) we have almost no hysteresis under these conditions. Below you will find some magnetoresistance data that we have just measured on another more reversible superconductor under similar conditions. No hysteresis can be seen here. So we can exclude experimental artifacts, but we believe that for a layered superconductor like $\text{Nb}_x\text{Bi}_2\text{Se}_3$ in parallel fields, any hysteresis in magnetostriction is most likely due to enhanced flux pinning when the flux has to enter or exit the sample and experiences the pinning from the different parallel layers.

In the revised manuscript we mention the slow sweep rate and exclude experimental artifacts as possible cause of the observed hysteresis to better substantiate our statement (page 5, line 3-8).

Comment 2:

2. Concerning the strain dependence of the nematic properties. There are two recent research those address this issue in AxBi_2Se_3 materials:

<https://arxiv.org/abs/1910.03252> and <https://doi.org/10.1103/PhysRevB.100.224509>

The authors should somehow comment on relation of their results with these studies.

Our response:

We thank the reviewer for this suggestion. We have added the two references and briefly compare their results with our own results (page 7, line 2-4). In both articles they demonstrate the control of nematic superconductivity by uniaxial strain, which is in perfect agreement with our observation that the nematic order causes a crystalline distortion. The latter implies that the nematic transition is strain dependent via the thermodynamic Ehrenfest or Clausius Clapeyron equation, as discussed in our manuscript.

Responses to Reviewer #3:

Comment 1:

[A] Concerning my comments 1 and [1](2) in the previous round, it would be better to include some figures in the response letter to the manuscript (probably in Supplemental Information); the one comparing raw $\Delta L/L_0$ data of Sample 2 and the empty dilatometer, and the one comparing measured thermal expansion coefficient (α) of silicon with the literature data. These two figures are quite convincing and should be shown to the readers.

Our response:

We thank the reviewer for this suggestion. The figures were included in the Supplementary Information as Supplementary Figure 7 and 8.

Comment 2:

[B] Concerning my comment [1](4), please add a short sentence on the sweep direction in Method.

Our response:

The direction and speed of the sweep was added on page 9, line 8, thank you!

Comment: 3

[C] Concerning my comment [9], I asked about the CALCULATED susceptibility, but the authors response was on the MEASURED susceptibility. I'm curious why there is a substantial anisotropy between χ_{xx}/χ_{yy} and χ_{zz} in Fig.5. It would be nice if the authors put a short explanation on this in the final version.

Our response:

We have added the following statement (page 6, line 28-31):

“While the in-plane anisotropy of the susceptibility is only finite below the nematic transition, the crystal symmetry allows χ_{zz} to be distinct already above T_{nem} . The magnitude of the out-of-plane anisotropy is determined by the ratio of the electron velocities in the corresponding directions.”

Comment 4:

[D] In the abstract (line 27), "partially melted superconductor" should be "partially melted superconductivity" or "partially melted superconducting order parameter".

Our response:

This has been corrected, thank you very much!

Comment 5:

[E] In the introduction (line 42), "Vestigial order, where the primary order is superconductivity, has not ..." should be "Vestigial order whose the primary order is superconductivity has not ...". Here, restrictive clause should be used. Thus, commas should be removed.

Our response:

This has been corrected, thank you very much!